# Binding and Action of Triphenylphosphonium Analog of Chloramphenicol upon the Bacterial Ribosome

**DOI:** 10.3390/antibiotics10040390

**Published:** 2021-04-05

**Authors:** Chih-Wei Chen, Julia A. Pavlova, Dmitrii A. Lukianov, Andrey G. Tereshchenkov, Gennady I. Makarov, Zimfira Z. Khairullina, Vadim N. Tashlitsky, Alena Paleskava, Andrey L. Konevega, Alexey A. Bogdanov, Ilya A. Osterman, Natalia V. Sumbatyan, Yury S. Polikanov

**Affiliations:** 1Department of Biological Sciences, University of Illinois at Chicago, Chicago, IL 60607, USA; cchen234@uic.edu; 2Department of Chemistry, Lomonosov Moscow State University, 119991 Moscow, Russia; julidev@yandex.ru (J.A.P.); zkh_msu@mail.ru (Z.Z.K.); tashlitsky@genebee.msu.ru (V.N.T.); bogdanov@genebee.msu.su (A.A.B.); 3Center of Life Sciences, Skolkovo Institute of Science and Technology, 143028 Skolkovo, Russia; dmitrii.lukianov@skoltech.ru; 4A.N. Belozersky Institute of Physico-Chemical Biology, Lomonosov Moscow State University, 119991 Moscow, Russia; tereshchenkov@list.ru; 5Laboratory of Multiscale Modeling of Multicomponent Materials, South Ural State University, 454080 Chelyabinsk, Russia; makarovgi@susu.ru; 6Petersburg Nuclear Physics Institute, National Research Center (NRC) “Kurchatov Institute”, 188300 Gatchina, Russia; polesskova_ev@pnpi.nrcki.ru (A.P.); konevega_al@pnpi.nrcki.ru (A.L.K.); 7Peter the Great St.Petersburg Polytechnic University, 195251 Saint Petersburg, Russia; 8National Research Center (NRC) “Kurchatov Institute”, 123182 Moscow, Russia; 9Department of Pharmaceutical Sciences, University of Illinois at Chicago, Chicago, IL 60607, USA; 10Center for Biomolecular Sciences, University of Illinois at Chicago, Chicago, IL 60607, USA

**Keywords:** chloramphenicol, antibiotic, 70S ribosome, X-ray structure, translation inhibitor, binding affinity, peptidyl transferase center, nascent peptide exit tunnel

## Abstract

Chloramphenicol (CHL) is a ribosome-targeting antibiotic that binds to the peptidyl transferase center (PTC) of the bacterial ribosome and inhibits peptide bond formation. As an approach for modifying and potentially improving the properties of this inhibitor, we explored ribosome binding and inhibitory properties of a semi-synthetic triphenylphosphonium analog of CHL—CAM-C4-TPP. Our data demonstrate that this compound exhibits a ~5-fold stronger affinity for the bacterial ribosome and higher potency as an in vitro protein synthesis inhibitor compared to CHL. The X-ray crystal structure of the *Thermus thermophilus* 70S ribosome in complex with CAM-C4-TPP reveals that, while its amphenicol moiety binds at the PTC in a fashion identical to CHL, the C4-TPP tail adopts an extended propeller-like conformation within the ribosome exit tunnel where it establishes multiple hydrophobic Van der Waals interactions with the rRNA. The synthesized compound represents a promising chemical scaffold for further development by medicinal chemists because it simultaneously targets the two key functional centers of the bacterial ribosome—PTC and peptide exit tunnel.

## 1. Introduction

The majority of antibiotics suppress pathogenic bacteria, and thereby cure infections, by selectively inhibiting their ribosomes—molecular machines that are responsible for protein biosynthesis. These ribosome-targeting antibiotics can hinder protein synthesis through binding at the functional centers of the ribosome. Chloramphenicol (CHL) is one of the oldest and the most well-studied ribosomal drugs that target the catalytic peptidyl transferase center (PTC) located at the heart of the large ribosomal subunit (50S) where peptide bond formation occurs [1]. As with many other PTC-targeting antibiotics, CHL binds to the ribosome in the A-site cleft, a crevice between the bases of nucleotides A2451 and C2452 of the 23S rRNA, where the aromatic nitrobenzyl ring of CHL establishes π–π stacking interactions with the nucleobases of A2451 and C2452 [2,3,4]. At this location, the ribosome-bound CHL competes with the amino acid side chains of the incoming aminoacyl-tRNAs (aa-tRNAs). Moreover, recent in vitro (toe-printing) and in vivo (Ribo-seq) studies showed that CHL does not act as a universal inhibitor of peptide bond formation and, instead, exhibits a context-specific mode of action inhibiting peptide bond formation only when the ribosome carries a nascent peptide with either alanine, serine, or threonine in the penultimate position [5]. These data also suggest that the ribosome-bound CHL molecule interacts not only with the ribosomal parts but also with the growing polypeptide chain located in the nascent peptide exit tunnel (NPET), through which the newly synthesized proteins exit the ribosome [1].

Although CHL specifically targets bacterial 70S ribosomes, it can also bind to mammalian mitochondrial ribosomes [6,7,8], causing major side effects [9,10]. As a result, the clinical usage of CHL is currently limited to developing countries where it is used against infections that cannot be eradicated using other treatments. Moreover, the alarming rise of antibiotic resistance among pathogenic bacteria continues to restrict the usage of the existing arsenal of antibiotics, creating a significant and ever-growing threat to healthcare [11]. As with other antibiotics, bacteria have developed multiple different resistance mechanisms against CHL, such as permeability barriers, drug efflux systems, as well as chemical modifications of the drug or its target, the 70S ribosome, that allow them to survive in the presence of CHL and even further reduce its medical utility [12]. One of the typical approaches to improve the target specificity of an antibiotic and to overcome resistance to it among pathogens is to change the existing drugs through rational drug design.

CHL has been frequently used as a platform to obtain derivatives with increased potency [12]. This drug is especially amenable to chemical derivatization because its dichloracetyl moiety can be easily replaced with a variety of other chemical scaffolds, such as amino acids [13], peptides [14], or other acyl-carrying groups [15], rendering it with new properties. For example, in one of the previous studies, CHL analogs with the dichloroacetyl moiety replaced by a polyamine extension that carries a positive charge and hydrophobic phenyl groups showed improved antibacterial and antitumor activities compared to CHL [15,16]. Moreover, recently, we have synthesized several amino acid analogs of CHL and demonstrated that the histidyl-derivative of CHL, which carries aromatic and positively charged side chain, exhibits substantially higher affinity for the bacterial ribosome due to strong stacking interactions of the histidyl moiety of this compound with the nucleotide U2506 of the 23S rRNA [13]. Although the higher affinity of this and other CHL amino acid derivatives did not correlate with their inhibitory properties, these new compounds exhibited different modes of action and context specificity compared to the original CHL [13]. Thus, synthesis of CHL derivatives through alteration(s) of its chemical structure might result in novel compounds showing unique interactions with the ribosome and potentially greater pharmaceutical properties.

In the current study, we designed, synthesized, and explored a semi-synthetic triphenylphosphonium (TPP) analog of CHL, CAM-C4-TPP, with the goal of obtaining a new group of CHL derivatives with potentially improved or even unique properties. To this end, the dichloromethyl group (of dichloroacetyl moiety) of the parent CHL compound was replaced by a butyl linker (C4) connected to the triphenylphosphonium residue (TPP), resulting in the CAM-C4-TPP molecule (Figure 1, compound **7**). Using competition binding assay, we show that CAM-C4-TPP exhibits stronger binding to the bacterial ribosome compared to CHL. Moreover, it displays a higher potency as an inhibitor of in vitro protein synthesis. We also showed that the mode of action of CAM-C4-TPP differs from the site-specific action of CHL. Finally, by determining the high-resolution X-ray crystal structure of the CAM-C4-TPP in complex with the *Thermus thermophilus* 70S ribosome, we found that the TPP moiety forms specific and peculiar stacking interactions with several nucleobases of the 23S rRNA, rationalizing its improved binding to the ribosome. To our knowledge, this type of interaction was not known for any of the previously published structures of ribosome-bound antibiotics. Therefore, our study provides a structural basis for further exploration of this new group of CHL-based derivatives by medicinal chemists in their effort to develop better PTC-targeting drugs. We believe that through an iterative search of the most optimal length of the linker connecting the CAM and the TPP moieties [17] or by combining the TPP moiety with other known ribosomal drugs, we potentially can create new antibiotics with a high affinity to the modified ribosomes from drug-resistant pathogens.

## 2. Results and Discussion

### 2.1. Synthesis of Triphenylphosphonium Chloramphenicol Analog

In this work, we set out to explore the properties of the triphenylphosphonium (TPP) analog of CHL—CAM-C4-TPP. The rationale for such a chemical scaffold came from our initial idea to create a hybrid molecule that would consist of two parts: (i) the amphenicol moiety anchoring it in the canonical CHL binding site within the PTC of the bacterial ribosome; and (ii) an additional NPET-binding group forming multiple interactions with the walls of NPET, which should be strong enough but non-specific so that interactions with various different nucleotides (lining up the NPET) would be possible, and (iii) the linker connecting the two endpieces. Such design of a target molecule would ensure its optimal binding to the ribosome and high tolerance to the length of the linker because similar non-specific interactions with the walls of NPET are expected to be possible at different NPET depths. Based on our initial *in silico* modeling as well as published structural and functional data on other CHL analogs [13,14,15], we decided to begin with a four-carbon linker (C4) for the synthesis of our target molecule (Figure 1). However, we realized that by using the linkers of various lengths, we could potentially create a compound that binds and inhibits the drug-resistant ribosomes, for example, of Cfr-expressing pathogens [18]. As an NPET-binding group, in this work, we have chosen TPP molecule because (i) its positive charge, which is delocalized over the relatively large hydrophobic surface of benzene rings, should provide non-specific interactions with the negatively charged phosphates of the 23S rRNA; and (ii) its three phenyl groups, which should be available for stacking with nucleobases. The TPP part itself belongs to a group of penetrating cations, which were first described more than 50 years ago [19]. The positive charge of the ionized phosphorus atom in TPP is scattered over a large hydrophobic molecule preventing the organization of water dipoles around this molecule that endows it with the ability to cross biomembranes using the energy of transmembrane potential [19]. By disrupting the membrane potential in a bacterial cell, TPP was previously shown to have antimicrobial effects [20,21,22]. Thus, by combining CHL and TPP groups in the same molecule, we could potentially create a superior and more efficient drug because of its dual mechanism of action.

The designed CAM-C4-TPP compound was synthesized through acylation of chloramphenicol amine (CAM) with carboxyl derivative of TPP using succinimide ester (Figure 1). The overall semi-synthesis scheme for CAM-C4-TPP includes three steps: (i) acid hydrolysis of CHL (**1**) to yield CAM (**2**) [23]; (ii) preparation of (4-carboxybutyl)triphenyl-phosphonium bromide (**5**) by condensation of 5-bromopentanoic acid (**3**) and triphenylphosphin (**4**); and (iii) acylation of CAM by succinimide ester of (4-carboxybutyl)triphenylphosphonium bromide (**6**). The chemical structure of the resulting CAM-C4-TPP molecule was confirmed by mass-spectrometric analysis as well as 1H-, 13C-, and 31P-NMR.

### 2.2. Protein Synthesis Is the Primary Target of CAM-C4-TPP Action In Vivo

Since CAM-C4-TPP is a semi-synthetic analog of the PTC-targeting antibiotic CHL, we expected that, similar to CHL, CAM-C4-TPP would also bind and act upon the bacterial ribosome inhibiting protein synthesis. However, we could not rule out the possibility that the presence of the additional TPP group could alter the intracellular target of the CAM-C4-TPP molecule. To check whether protein synthesis remains the primary target of CAM-C4-TPP action in the bacterial cells, we used an *E. coli*-based in vivo reporter system that is designed for screening of inhibitors targeting either DNA replication or protein synthesis [24]. In this assay, sub-inhibitory concentrations of antibacterial compounds that stall translation (e.g., CHL, or erythromycin) induce expression of far-red fluorescent protein reporter Katushka2S (Figure 2A, CHL and ERY, red pseudocolor rings). Compounds that trigger SOS response, such as inhibitors of DNA gyrase (e.g., levofloxacin), induce expression of Red Fluorescent Protein (RFP) reporter (Figure 2A, LEV, green pseudocolor ring). Consistent with the previous studies showing that removal of the dichloroacetyl group from CHL and turning it into chloramphenicol amine (CAM) renders it inactive [23] due to the loss of the critical interactions with the ribosome [2], we observed no induction of any of the two reporters by CAM (Figure 2A, CAM, no colored rings) as well as no cell growth inhibition (Figure 2A, CAM, no dark zone in the middle). Interestingly, the C4-TPP tail by itself inhibits bacterial cell growth (Figure 2A, C4-TPP, dark area in the middle) but does not induce any of the two reporters, suggesting that, in the absence of the amphenicol moiety, it likely acts via a completely different intracellular target (e.g., membrane [20,21,22]). When *E. coli* cells carrying the reporter were exposed to CAM-C4-TPP, the expression of the fluorescent protein Katushka2S, but not of RFP, was strongly activated, indicating that the drug specifically inhibits protein synthesis (Figure 2A, CAM-C4-TPP, red pseudocolor ring).

### 2.3. CAM-C4-TPP Binds Tightly to the Bacterial Ribosome and Inhibits Protein Synthesis

To assess whether the synthesized CAM-C4-TPP compound exhibits a higher affinity for the bacterial 70S ribosome due to the presence of the hydrophobic and charged TPP tail, we used the competition-binding assay exploiting either BODIPY-labeled CHL (BODIPY-CAM, Appendix A) or erythromycin (BODIPY-ERY) [13,25,26]. The amphenicol moieties of CAM-C4-TPP, its parent CHL, and the fluorescent derivative BODIPY-CAM are expected to have identical ribosome binding sites and, therefore, these compounds are expected to compete with each other for binding to the ribosome. Although CHL binds in the A site of the PTC [2,3,4,27] and ERY binds in the upper part of the NPET [2,3,28], their binding sites sufficiently overlap resulting in competition for binding to the ribosome between CHL- and ERY-based compounds [2,29]. The apparent dissociation constants *K_Dapp_* of CHL determined from the displacement of BODIPY-CAM (2.6 ± 1.5 µM) or BODIPY-ERY (2.8 ± 0.5 µM) are consistent with each other and with the previously published data [13], including data determined by direct [^14^C]-CHL binding (2.3 µM [30]). Using this approach, we found that CAM-C4-TPP exhibits a considerably higher (~5-fold) affinity for the ribosome compared to parent CHL in both displacement assays with BODIPY-CAM (*K_Dapp_* = 0.6 ± 0.4 µM for CAM-C4-TPP vs. 2.6 ± 1.5 µM for CHL) or BODIPY-ERY (*K_Dapp_* = 0.61 ± 0.07 µM for CAM-C4-TPP vs. 2.8 ± 0.5 µM for CHL) (Figure 2B). Apparently, the positive charge of the TPP substituent contributes to the more efficient ribosome binding of CAM-C4-TPP as previously observed for the positively charged amino acid derivatives of CHL [13]. The presence of hydrophobic TPP phenyl rings, which could potentially stack onto the nucleobases lining up the NPET, might also positively affect the overall affinity of CAM-C4-TPP for the 70S ribosome. Consistent with our design idea, these data suggest that replacement of the dichloroacetyl moiety of CHL with the C4-TPP tail results in a 5-fold more efficient binding of the resulting CAM-C4-TPP compound to the 70S ribosome.

To check whether the improved binding of CAM-C4-TPP to the ribosome correlates with stronger inhibition of translation, we tested its ability to interfere with in vitro protein synthesis. Addition of either CHL or CAM-C4-TPP to the cell-free transcription-translation system based on the *E. coli* S30 extract resulted in a dose-dependent inhibition of the synthesis of firefly luciferase reporter by both drugs (Figure 2D). In these experiments, CAM-C4-TPP exhibited more than 3-fold better IC_50_ (0.9 ± 0.2 µM) compared to that of CHL (2.8 ± 0.5 µM) with nearly complete inhibition of protein synthesis observed at ~10 µM for each of the drugs. This data suggest that turning the CHL molecule into CAM-C4-TPP endows it with superior translation inhibitory properties. Thus, CAM-C4-TPP provides a promising foundation for future ribosome-targeting drug development.

### 2.4. TPP Moiety of CAM-C4-TPP Establishes Unique Interactions with the Ribosome

To understand the functional relevance of the introduced TPP moiety of CAM-C4-TPP (Figure 3A) in the context of the bacterial ribosome, we have determined the crystal structure of CAM-C4-TPP bound to the *Thermus thermophilus* (*Tth*) 70S ribosome at 2.80Å resolution (Appendix A). In this experiment, we used *Tth* 70S ribosomes complexed with *E. coli* protein Y (PY) as a tool to obtain structures of higher resolution [13,28,31,32]. This approach is based on our previous finding that binding of PY to a vacant 70S ribosome stabilizes it by locking the head of the 30S subunit in an unrotated state, which leads to overall better diffraction and, subsequently, improved structural resolution [31,32]. Since the binding site of PY is located on the small ribosomal subunit, where it overlaps with mRNA and all three tRNAs, it does not interfere with the binding of CHL or its analogs in the PTC of the large ribosomal subunit. As a result, by using ribosome-PY complexes, we were able to obtain higher resolution and overall better quality electron density maps of the ribosome-bound CAM-C4-TPP drug than it was possible otherwise. The unbiased difference Fourier map for CAM-C4-TPP, calculated using the observed amplitudes from the crystal and the amplitudes and phases derived from a model of the antibiotic-free ribosome, revealed unique positive electron density peaks (Figure 3B,C) resembling characteristic features of the CAM-C4-TPP chemical structure (Figure 3A). A single binding site spanning both PTC and NPET of the large ribosomal subunit was observed for the CAM-C4-TPP (Figure 4A).

The overall binding position of the amphenicol moiety of ribosome-bound CAM-C4-TPP is identical to those observed previously for the parent CHL compound (in complex with either *Escherichia coli* [4] or *Tth* [3] 70S ribosomes in the absence of mRNA and tRNAs or in the complex with mRNA and all three tRNAs (Figure 4B, green; Appendix A) [2]) as well as for the CHL analog histidyl-CAM (Figure 4B, blue) [13]. In this binding site, the aromatic nitrobenzyl ring of the drug intercalates into the A-site cleft, where it is sandwiched between the nucleobases A2451 and C2452 of the 23S rRNA and forms π–π stacking interactions with them (Figure 4C,D). Similar to CHL, the amphenicol moiety of CAM-C4-TPP also forms a hydrogen bond (H-bond) with the phosphate of U2504 residue (Figure 4C). Consistent with the drug design rationale, our structure confirms that the amphenicol part of the CAM-C4-TPP molecule anchors it in the PTC of the 70S ribosome, directing the attached C4-TPP tail towards the macrolide binding site in the NPET.

The overall quality of the obtained electron density map allowed us to unambiguously model all parts of the positively charged and hydrophobic C4-TPP tail (Figure 3B,C). The observed location of this tail within the ribosome is the most exciting finding of the current study. Although this moiety is likely to be extremely flexible in solution, once bound to the bacterial ribosome, the C4-TPP tail extends into the NPET of the large ribosomal subunit where it adopts a unique conformation resembling the shape of a toilet plunger (or an airplane propeller with a handle). The observed tight coordination of the C4-TPP tail is achieved through multiple hydrophobic Van der Waals interactions of each of the three phenyl rings with several nucleotides lining up the NPET (Figure 4C,D and Appendix A). For example, the phenyl moieties of the TPP group form π–π stacking interactions with nucleotides A2062 (edge-to-face; Appendix A), G2505 (edge-to-face Appendix A), and C2610 (parallel-displaced; Appendix A) as well as Van der Waals contacts with nucleotides A2057, A2058, A2059, and U2611 of the 23S rRNA (Appendix A). Moreover, similar to the carbonyl group of the dichloroacetic moiety of CHL, the carbonyl group of the C4 linker provides additional stabilization to the C4-TPP tail via the formation of the H-bond with the N6 atom of nucleotide A2062 of the 23S rRNA. This interaction results in rotation of the nucleobase of A2062 by ~160° around its N-glycosidic bond into a position where it forms a Hoogsteen base pair with the residue m^2^A2503 (Figure 4C). Similar re-orientation of this nucleotide was observed previously in the ribosome-CHL structures [2,3,4], as well as in structures of ribosome-bound amino acid derivatives of CHL [13] and macrolides [2,28]. These additional interactions of the TPP moiety of the CAM-C4-TPP likely account for its increased affinity to the ribosome and better inhibitory properties compared to those of CHL.

Resistance to many PTC-acting inhibitors (including CHL) can be conferred by Cfr methyltransferase that methylates residue A2503 of the 23S rRNA at the C8 atom [18]. Assuming that the placement of the modified A2503 nucleotide within the ribosome remains unchanged, the methyl group added to the A2503 by Cfr would overlap with the CHL binding pocket resulting in its displacement from the ribosome (Appendix A). In silico modeling of the C8-methylated residue A2503 in our structure of ribosome-bound CAM-C4-TPP shows that the sterical clash between the drug, whose placement in the PTC is very similar to CHL, and the methyl group is somewhat less pronounced compared to CHL (Appendix A) due to a more remote position of the C4-TPP tail of CAM-C4-TPP relative to the methylation site. Although these rather small differences are unlikely to render CAM-C4-TPP active against Cfr-expressing pathogens, this new type of CHL derivatives could be a promising new scaffold for the potential development of the new analogs that can counteract the Cfr-mediated resistance mechanism. Further experiments would be required to check whether CAM-C4-TPP or its chemical ‘brothers’ (with longer or shorter linkers between the amphenicol and the TPP moieties) are active against Cfr-modified drug-resistant ribosomes.

Several different classes of ribosome-targeting antibiotics, such as macrolides, lincosamides, and type B streptogramins, bind and act in the NPET of the bacterial ribosome. Superposition of the structure of CAM-C4-TPP in complex with the 70S ribosome with the known structures of ribosome-bound macrolide ERY (Appendix A), lincosamide clindamycin (CLI, Appendix A), or type B streptogramin quinupristin (QIN, Appendix A) shows that their binding sites largely overlap with each other. In contrast to the parent molecule CHL, whose binding site spans only the PTC (Figure 5B), the C4-TPP tail of CAM-C4-TPP extends into the ribosomal exit tunnel and partially obstructs it (Figure 5C). This is in striking similarity to chemically unrelated macrolides (Figure 5D), as well as lincosamides (Figure 5E) and streptogramins B (Figure 5F), which only partially occlude the exit tunnel, still allowing for many of the experimentally tested nascent peptides to pass through [33,34,35]. Our structural analysis reveals that, in the presence of ribosome-bound CAM-C4-TPP, there is still enough space in the ribosomal exit tunnel to accommodate a nascent peptide. This suggests that CAM-C4-TPP should directly interact with the nascent peptides during their synthesis and, therefore, is likely to exhibit a context-specific mode of action.

### 2.5. CAM-C4-TPP Shows Unique Context-Specificity of Action

The inhibition of protein synthesis by the PTC-targeting antibiotics depends on the amino-acid sequence of the nascent polypeptide chain and the identity of the incoming aminoacyl-tRNA in the A site of the ribosome [5]. For example, CHL arrests translation only when alanine, and to a lesser extent, serine or threonine appear in the penultimate position (E site) of the growing polypeptide chain and only if there is no glycyl-tRNA in the A site of the ribosome [5]. However, the context-specificity of drug-induced ribosome stalling was not observed for the recently studied amino acid analogs of CHL, suggesting that the site-specificity of CHL action critically depends on the presence of the dichloroacetyl moiety that was replaced with histidyl or lysyl residues in CHL amino acid analogs [13]. To check whether or not the context-specificity of action of CAM-C4-TPP, which showed higher affinity to the ribosome (Figure 2B,C) and exhibited better inhibitory activity in the cell-free translation system (Figure 2D), is similar to that of the parent compound CHL, we used primer extension inhibition assay (toe-printing). This method allows unambiguous identification of the drug-induced ribosome stalling site along the mRNA with a single nucleotide precision [36,37]. This technique also allows for the determination of the context-specificity of drug action [5]. A large set of ribosome-targeting antibiotics has been previously tested by toe-printing using *rst1* or *rst2* mRNAs as templates [37], because their sequences contain codons for almost all proteinogenic amino acids. Therefore, in order to be able to directly compare our results for CAM-C4-TPP with those for other antibiotics, we have chosen exactly the same experimental system.

**Figure 5 antibiotics-10-00390-f005:**
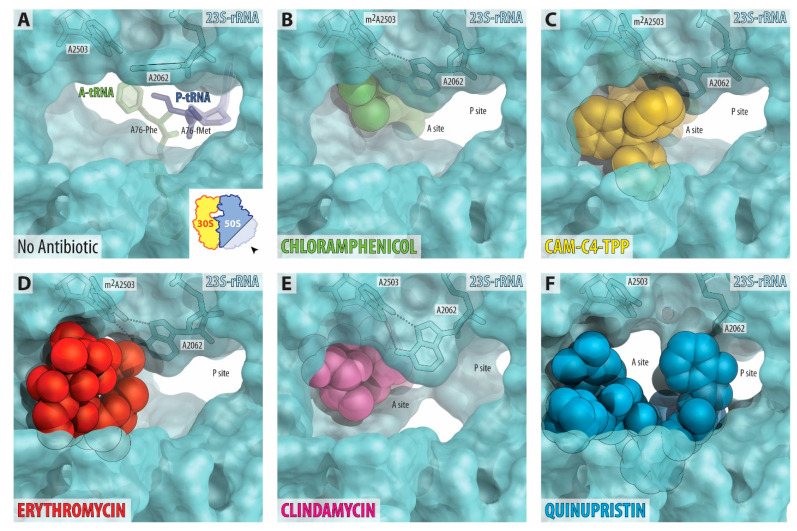
Comparison of the binding sites of several antibiotics in the nascent peptide exit tunnel. (**A**) The lumen of the NPET in the drug-free 70S ribosome (PDB entry 4Y4P [32]). The view is from the wide-open part of the tunnel onto the PTC, as indicated by the inset. Aminoacylated nucleotides A76 of the A- and P-site tRNA are shown in green and dark blue, respectively. Note that nucleotide A2062 of the 23S rRNA is pointed toward the viewer and is not involved in Hoogsteen-edge base pairing with the nucleotide m^2^A2503 in the NPET wall. (**B**–**F**) Occlusion of the nascent peptide exit tunnel by chloramphenicol (**B**, green, PDB entry 6ND5 [2]), CAM-C4-TPP (**C**, yellow), macrolide erythromycin (**D**, red, PDB entry 6XHX [28]), lincosamide clindamycin (**E**, magenta, PDB entry 4V7V [4]), and type B streptogramin quinupristin (**F**, blue, PDB entry 4U26 [38]). Note that, unlike A-site binding chloramphenicol, the NPET-binding antibiotics, such as macrolides, lincosamides, or streptogramins B, as well as CAM-C4-TPP partially obstruct the lumen of the NPET. Also, note that, in the presence of ribosome-bound CAM-C4-TPP, there is enough space for a nascent peptide to pass through.

As expected, the addition of CHL to the cell-free transcription-translation system programmed with *rst1* or *rst2* mRNAs resulted in a predominant ribosome stalling when the 6th serine (Ser6) or 4th alanine (Ala4) residues appeared in the penultimate position of the growing polypeptide chains, respectively (Figure 6A,B, lanes 2 and 5, red and navy arrowheads). However, unlike CHL that stalls ribosomes at various Ala/Ser/Thr codons inside the open reading frames (ORFs), CAM-C4-TPP also causes a significant portion of the ribosomes to stall at the initiator codons (Figure 6A,B, lanes 3 and 6, black arrowheads). Nevertheless, in the presence of CAM-C4-TPP, a fraction of ribosomes manages to escape the start codon, continues translation, and stalls at later codons within the ORF (Figure 6A,B, lanes 3 and 6). While some of these drug-induced ribosome stalling sites are identical between CHL and CAM-C4-TPP, there are few CAM-C4-TPP-specific stalling positions suggesting that it has an idiosyncratic mode of action and its unique context-specificity, which is different from that of the parent CHL.

Overall, our toe-printing experiments revealed that, unlike CHL, which interferes with the binding of aa-tRNAs during elongation cycles, CAM-C4-TPP could stall ribosomes at the start codon during the initiation step of protein synthesis. This is likely due to the observed higher affinity of CAM-C4-TPP to the ribosome (Figure 2B,C) and the fact that it was added to the toe-printing mixture prior to the reaction onset. Moreover, this observation is consistent with our structural data showing that the TPP moiety of CAM-C4-TPP extends into the NPET, where it forms a compound-specific network of interactions with the nucleotides of the 23S rRNA. Unlike CHL, such enhanced interactions of CAM-C4-TPP with the ribosome and its overall better anchoring in the NPET are likely to effectively outcompete even the first incoming aa-tRNA enabling CAM-C4-TPP to act even during the first elongation cycle. This is in contrast to the general mechanism of action of CHL, which requires a short nascent peptide to be synthesized first before it can cause ribosome stalling at Ala/Ser/Thr codons of the ORF. The observed CAM-C4-TPP-induced ribosome stalling at the downstream codons along the mRNA (Figure 6A,B, lanes 3 and 6, colored arrowheads) indicates that CAM-C4-TPP also interacts with the ribosome during translation elongation, which is consistent with our structural data showing that CAM-C4-TPP does not obstruct exit tunnel completely, leaving some space for the nascent peptide to fit (Figire 5C). It is tempting to speculate that the ability of CAM-C4-TPP to interfere with the first round of translation makes it a more potent translation inhibitor compared to CHL.

## 3. Materials and Methods

### 3.1. Reagents

The following reagents were used: chloramphenicol (Sigma, Steinheim, China); 1-hydroxysuccinimide, *N,N’*-dicyclohexylcarbodiimide, triphenylphosphine, and 5-bromopentanoic acid (Sigma-Aldrich, Steinheim, Germany); *N*-hydroxysuccinimide ester of BODIPY FL-C5 (Invitrogen, Eugene, OR, USA). Fluorescent ERY (BODIPY-ERY) and CHL (BODIPY-CAM) derivatives were synthesized as described previously [25,39]. The C4-TPP compound was obtained using previously published procedure [20].

### 3.2. Chemical Synthesis of CAM-C4-TPP

The general scheme for the synthesis of CAM-C4-TPP is shown in Figure 1, and the details of chemical synthesis are provided in the Appendix A. Chloramphenicol amine (CAM, **2**, (1R,2R)-2-amino-1-(4-nitrophenyl)propane-1,3-diol) was prepared via acid hydrolysis of chloramphenicol (CHL, **1**) according to the previously published procedure [23], which was also used in our recent studies [13,26]. (4-Carboxybutyl)triphenyl-phosphonium bromide (C4-TPP, **5**) was obtained by condensation of 5-bromopentanoic acid (**3**) and triphenylphosphin (**4**) for 12 h at 85 °C. The resulting carbonic acid **5** was converted to the succinimide ester by reaction with 1-hydroxysuccinimide in the presence of *N,N’*-dicyclohexylcarbodiimide at 0 °C, which was further used for the acylation of CAM in the presence of diisopropylethylamine at room temperature. The final CAM-C4-TPP compound (**7**) was purified by column chromatography on silica gel. The purity and chemical structure of the obtained compound was confirmed by HPLC, LC-MS, and NMR spectroscopy (see Appendix A).

### 3.3. In Vivo Detection of Translation Inhibitors Using Pdualrep2 Reporter Strain

For the in vivo bioactivity test (Figure 2A), we used the reporter strain BW25113ΔtolC-pDualrep2 as described previously [24,40]. Briefly, 1.5 µL of the solutions of CAM-C4-TPP (20 mM), C4-TPP (20 mM), CAM (2 mM), and 1 µL of the solutions of CHL (2 mM), erythromycin (ERY, 6.8 mM), and levofloxacin (LEV, 69 nM) in DMSO were applied onto the agar plate that already contained a lawn of the reporter strain. After overnight incubation at 37 °C, the plate was scanned by ChemiDoc (Bio-Rad, USA) using “Cy3-blot” mode for RFP fluorescence and “Cy5-blot” mode for Katushka2S fluorescence.

### 3.4. In Vitro Binding Assay

The apparent dissociation constant of BODIPY-CAM (*K_Dapp_* = 3.5 ± 0.9 µM) was determined by its direct equilibrium binding to *E. coli* 70S ribosomes (Appendix A). A solution of BODIPY-CAM (4 nM) in the buffer containing 20 mM HEPES-KOH (pH 7.5), 50 mM NH_4_Cl, 10 mM Mg(CH_3_COO)_2_, 4 mM β-mercaptoethanol, and 0.05% Tween-20 was mixed with ribosomes in the range of concentrations from 5 nM to 5 μM, incubated for 30 min, and the values of fluorescence anisotropy were measured.

Binding affinities of CHL analog CAM-C4-TPP, as well as other tested compounds, for the *E. coli* ribosome (Figure 2B,C) were analyzed by a competition-binding assay using fluorescently labeled BODIPY-CAM or BODIPY-ERY as described before [13,25,26,41]. BODIPY-ERY (16 nM) or BODIPY-CAM (30 nM) were incubated with ribosomes (50 nM or 2.9 µM respectively) for 30 min at 25 °C in the buffer containing 20 mM HEPES-KOH (pH 7.5), 50 mM NH_4_Cl, 10 mM Mg(CH_3_COO)_2_, 4 mM β-mercaptoethanol, and 0.05% Tween-20.

Solutions of CHL, CAM, CAM-C4-TPP, or C4-TPP in different concentration ranges were added to the pre-formed ribosome/BODIPY-CAM or ribosome/BODIPY-ERY complexes. The mixtures were incubated for 2 h until equilibrium was reached and the values of fluorescence anisotropy were measured. The apparent dissociation constants were calculated for each tested compound based on the assumption that the competitive equilibrium binding of two ligands occurs at a single binding site as described in [42].

### 3.5. In Vitro Translation Inhibition Assays

In vitro assessment of inhibition of synthesis of firefly luciferase by the tested compounds (Figure 2D) was accomplished as described previously [43]. Briefly, the in vitro transcribed firefly luciferase mRNA was translated using *E. coli* S30 Extract System for Linear Templates (Promega, USA). Reactions containing 100 ng of mRNA and 5 µL of 0.1 mM of D-luciferin from the Steady-Glo Luciferase Assay System (Promega) were preincubated for 5 min with different concentrations (0.5 µM, 1 µM, 5 µM, 10 µM, 30 µM, 50 µM, 100 µM) of the tested compounds and carried out in 5-µL aliquots at 37 °C for 30 min. The activity of in vitro synthesized luciferase was measured by VICTOR X5 Multilabel Plate Reader (Perkin Elmer, USA) every 30 s. Calculations of the IC_50_ values for the tested compounds were carried out using GraphPad Prism software.

### 3.6. Toe-Printing Analysis

Assessment of the inhibition of primer extension (toe-printing) by the tested compounds (Figure 5) was carried out using Rst1 and Rst2 mRNA templates as previously described [37]. The concentrations of the tested compounds were 30 µM.

### 3.7. Crystallographic Structure Determination

70S ribosomes from *Thermus thermophilus* (strain HB8) were prepared as described previously [32]. Ribosome complexes with *Escherichia coli* protein Y were formed as described previously [31,32] in the buffer containing 5 mM HEPES-KOH (pH 7.6), 50 mM KCl, 10 mM NH_4_Cl, and 10 mM Mg(CH_3_COO)_2_, and then crystallized in the buffer containing 100 mM Tris-HCl (pH 7.6), 2.9% (w/v) PEG-20K, 9–10% (v/v) MPD, 175 mM arginine, 0.5 mM β-mercaptoethanol [32,43]. CAM-C4-TPP was added to the pre-formed ribosome complex to a final concentration of 400 µM prior to crystallization. Crystals were grown by the vapor diffusion method in sitting drops at 19 °C and stabilized as described previously [32], with CAM-C4-TPP being added to the stabilization buffers (250 µM).

Collection and processing of the X-ray diffraction data, model building, and structure refinement were performed as described in our previous publications [2,13,28,32]. Structural models and restraints for CAM-C4-TPP were generated using PRODRG online software (http://prodrg1.dyndns.org) [44]. The statistics of data collection and refinement are compiled in Appendix A. All figures showing atomic models were generated using the PyMol software (www.pymol.org).

## 4. Conclusions

The goal of this study was to synthesize the first triphenylphosphonium (TPP) derivative of CHL and explore the possibility of its usage as a new type of translation inhibitor with the idea to potentially develop it into a new antibiotic. We used semi-synthesis approach to generate the first TPP analog of CHL, CAM-C4-TPP, and examined its ribosome binding and inhibitory properties. Consistent with the chemical design, we have demonstrated that CAM-C4-TPP manifests antibacterial activity by targeting protein synthesis in vivo and exhibits ~5 times higher affinity for the bacterial ribosome as compared to the parent compound CHL. Moreover, we found that CAM-C4-TPP is a more potent inhibitor of protein synthesis in vitro with 3-fold better IC_50_ compared to that of CHL. Our crystal structure reveals CAM-C4-TPP binding site spanning both the PTC and the NPET, in which it adopts an extended propeller-like conformation and establishes compound-specific interactions with the nucleotides of the 23S rRNA at the PTC and NPET. The unique contacts of the TPP moiety with the ribosome open the possibility of increasing the selectivity of this class of translation inhibitors and possibly diminish their side effects mediated by the action upon mitochondrial translation. We hope that our findings, including the structure of the ribosome/CAM-C4-TPP complex reported here, will serve as a starting point for further exploration of this new class of ribosome-targeting inhibitors by medicinal chemists.

## Figures and Tables

**Figure 1 antibiotics-10-00390-f001:**
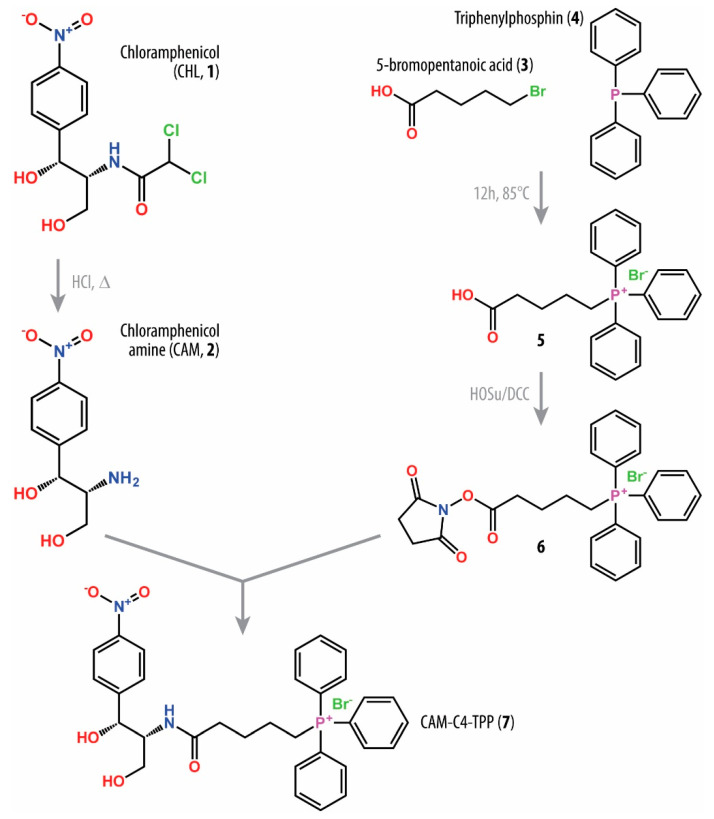
Schematic diagram of chemical synthesis of the triphenylphosphonium chloramphenicol analog CAM-C4-TPP. Each compound is designated by a number: (**1**) chloramphenicol (CHL); (**2**) chloramphenicol amine (CAM); (**3**) 5-bromopentanoic acid; (**4**) triphenylphosphin; (**5**) (4-carboxybutyl)triphenyl-phosphonium bromide; (**6**) (4-(*N*-succinimidyl)carboxybutyl)triphenyl-phosphonium bromide; (**7**) CAM-C4-TPP.

**Figure 2 antibiotics-10-00390-f002:**
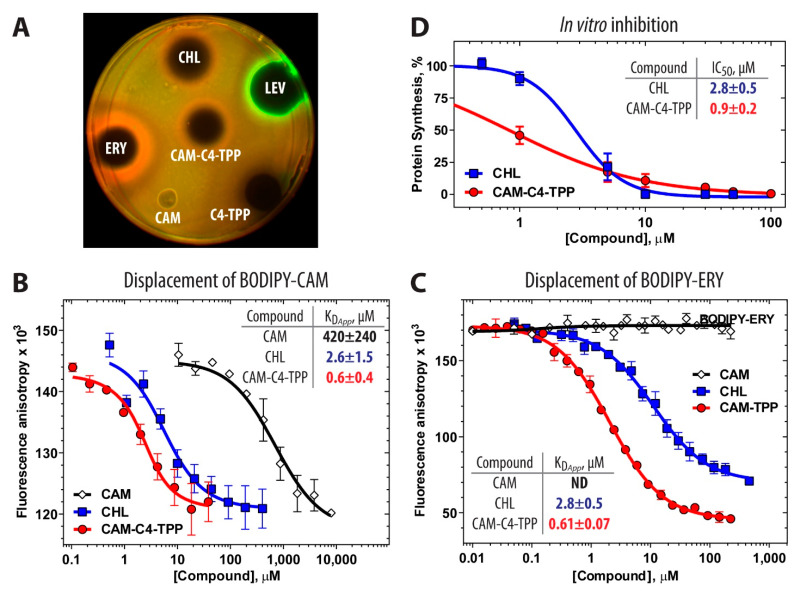
CAM-C4-TPP is a potent inhibitor of protein synthesis both in vivo and in vitro. (**A**) In vivo testing of CAM-C4-TPP activity using *E. coli* BW25113 ΔtolC pDualRep2 reporter strains. The induction of the red fluorescent protein (RFP) expression (green halo around the inhibition zone) is triggered by DNA-damage, while the induction of Katushka2S protein (red halo) occurs in response to ribosome stalling. Levofloxacin (LEV), erythromycin (ERY), and chloramphenicol (CHL) are used as controls. This image is an overlay of the two images from the independent detections of RFP (Cy3 mode) or Ktushka2S (Cy5 mode) by ChemiDoc MP (Bio-Rad) with RFP fluorescence shown in green pseudocolor and Katushka2S fluorescence shown in red pseudocolor. (**B**,**C**) Competition-binding assay to test the displacement of fluorescently labeled BODIPY-CAM (**B**) or BODIPY-ERY (**C**) analogs from the *E. coli* 70S ribosomes in the presence of increasing concentrations of CAM (open diamonds), CHL (blue squares), or CAM-C4-TPP (red circles) measured by fluorescence anisotropy. All reactions were repeated three times. Error bars represent standard deviation. ND, no displacement. The resulting values for apparent dissociation constants (K_D*app*_) are shown on each plot. (**D**) Inhibition of protein synthesis by increasing concentrations of CHL (blue squares) or CAM-C4-TPP (red circles) in the in vitro cell-free transcription-translation coupled system. The relative enzymatic activity of in vitro synthesized firefly luciferase is shown. Error-bars represent standard deviations of the mean of three independent measurements.

**Figure 3 antibiotics-10-00390-f003:**
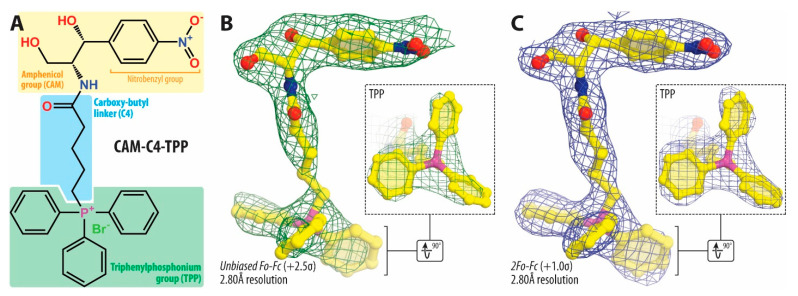
Chemical structure and electron density maps of ribosome-bound CAM-C4-TPP. (**A**) Chemical structure of CAM-C4-TPP compound. (**B**,**C**) Unbiased *Fo-Fc* (**B**) and 2*Fo-Fc* (**C**) electron density maps of CAM-C4-TPP in complex with the *T. thermophilus* 70S ribosome (green and blue mesh, respectively). The refined model of CAM-C4-TPP is displayed in its respective electron density before (**B**) and after (**C**) the refinement contoured at 2.5σ and 1.0σ, respectively. Carbon atoms are colored yellow, nitrogen atoms are blue, oxygen atoms are red, phosphorus is magenta. Note that the locations of the C4-linker, as well as TPP moiety, can be unambiguously determined from the electron density map.

**Figure 4 antibiotics-10-00390-f004:**
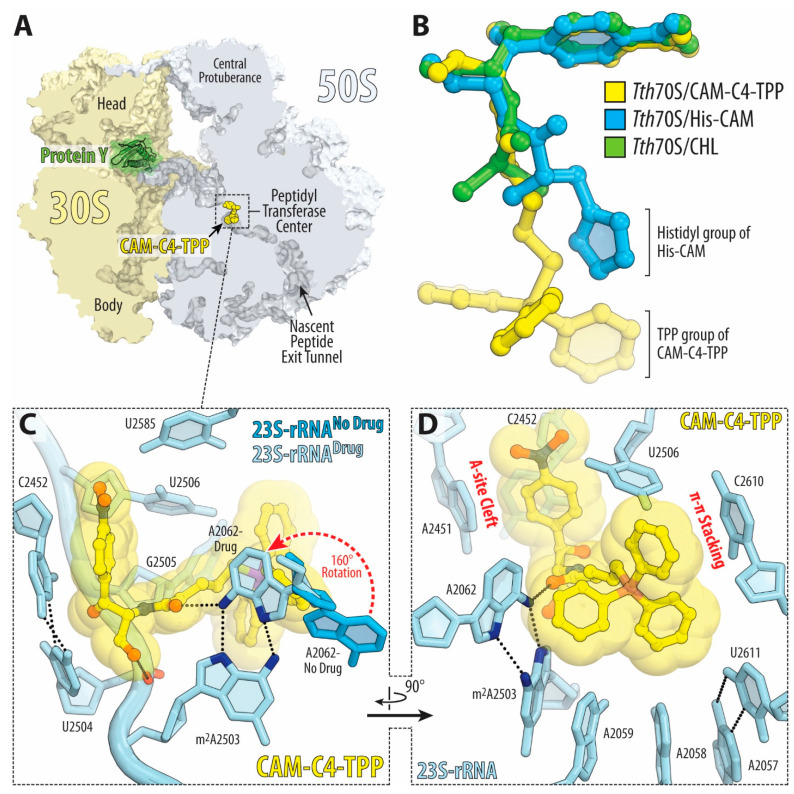
Structure of CAM-C4-TPP in complex with the 70S ribosome. (**A**) Overview of the drug-binding site (yellow) in the *T. thermophilus* 70S ribosome viewed as a cross-cut section through the nascent peptide exit tunnel. The 30S subunit is shown in light yellow, the 50S subunit is in light blue, ribosome-bound protein Y is colored green. (**B**) Superposition of the ribosome-bound CAM-C4-TPP (yellow) with the previous structures of CHL (green, PDB entry 6ND5 [2]) or CHL analog histidyl-CAM (blue, PDB entry 6CFJ [13]). All structures were aligned based on domain V of the 23S rRNA. (**C**,**D**) Close-up views of the CHL analog CAM-C4-TPP bound in the PTC and NPET of the 70S ribosome (*E. coli* numbering of the 23S rRNA nucleotides is used). Potential H-bond interactions are indicated with dashed lines. Note that by forming an H-bond with the base of nucleotide A2062 of the 23S rRNA (light blue) CAM-C4-TPP causes characteristic rotation of this nucleotide by approximately 160 degrees to form Hoogsteen base-pair with the m^2^A2503 of the 23S rRNA (red dashed arrow). N6 and N7 atoms of nucleotides A2062 and m^2^A2503 are highlighted in dark blue. The unrotated conformation of A2062 observed in the absence of the drug is shown in blue (PDB entry 4Y4P [32]).

**Figure 6 antibiotics-10-00390-f006:**
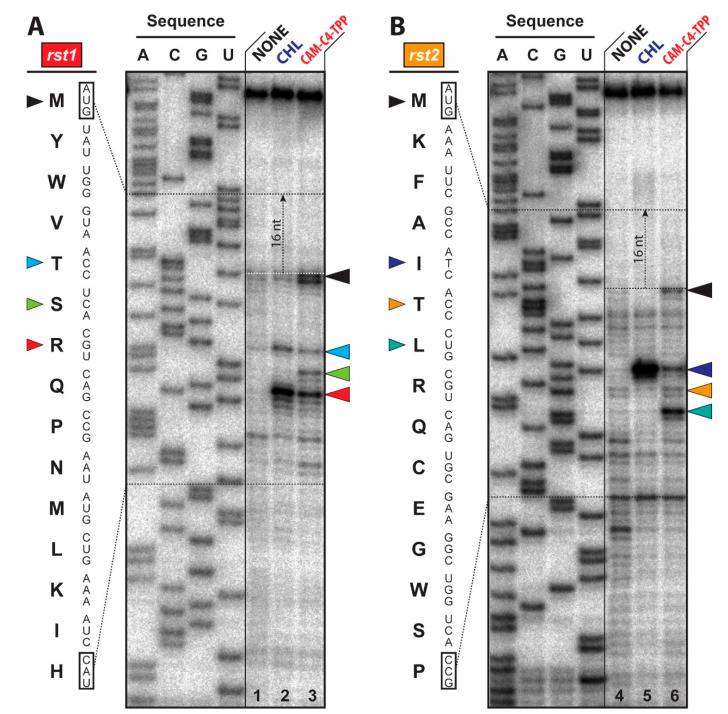
CAM-C4-TPP is capable of arresting translation at the start codon. Ribosome stalling by CAM-C4-TPP on *rst1* (**A**) and *rst2* (**B**) mRNAs in comparison with CHL, as revealed by reverse-transcription primer-extension inhibition (toe-printing) assay in a cell-free translation system. Nucleotide sequences of *rst1* and *rst2* mRNAs and their corresponding amino acid sequences are shown on the left. Black arrowheads mark translation arrest at the start codon, while variously colored arrowheads point to the drug-induced arrest sites within the coding sequences of each of the two used mRNAs. Note that, due to the large size of the ribosome, the reverse transcriptase used in the toe-printing assay stops 16 nucleotides downstream of the codon located in the P-site.

## Data Availability

Coordinates and structure factors were deposited in the RCSB Protein Data Bank with accession code 7MD7 for the *Tth* 70S ribosome in complex with CAM-C4-TPP and protein Y.

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
