# Peer review of "Binding and Action of Triphenylphosphonium Analog of Chloramphenicol upon the Bacterial Ribosome"

_antibiotics, 2021, doi:10.3390/antibiotics10040390_

Round 1
Reviewer 1 Report
In this study Chen and colleagues characterise the mode of action of a chloramphenicol derivative, CAM-C4-TPP, using a powerful combination of x-ray cystography, biochemistry and (a tiny bit) microbiology. Both X-ray and biochemical results are conclusive and presented well.
Derivatisation of CAM is a promising strategy for giving this old drug new properties and, eventually, bringing it back into the fight to with pathogens. For this microbiological characterisation of the drug is essential – after all, the drug will be used to fight bacteria. Therefore, I feel we are missing the in vivo data (mind you - I might be missing things altogether, it could be that all of the information of interest is already reported; I hope the authors excuse me if this is the case… ).
Would be good to see the following:
- Bacteriostatic and bactericidal activity on a couple of relevant target species, direct comparison with parental compound
- Toxicity on human cells (with a focus on mitochondrial translation)
In case these results were already reported before, please make the comparison in the introduction. If these results are to be reported elsewhere, please state that explicitly as a limitation of the current study. If some of these data can be generated for revision – this would be the best; amend the manuscript accordingly.
Specific comments:
- 2: ‘The majority of antibiotics kill pathogenic bacteria, and thereby cure infections, by selectively inhibiting their ribosomes – molecular machines that are responsible for protein biosynthesis.’ I would amend the wording: most of the translation-targeting antibiotics are bacteriostatic, not bactericidal.
- 6: ‘Protein synthesis is the primary target of CAM-C4-TPP action in vivo.’ The authors do not exhaustively investigate the possible off-targets effects their CAM-C4-TPP. One would need to do:
- microscopy (to assess membrane permeability and energetic status)
- metabolic labelling with both S35Met (translation) and radiolabelled U and T (transcription and replication).
Additionally, it would be really instructive to assess the bactericidal vs bacteriostatic effects of CAM vs CAM-C4-TPP (dose-dependently, since at higher concentrations CAM goes from bacteriostatic to bactericidal; I wonder how it goes for CAM-C4-TPP…).
However, all this might be too much to ask… I would, however, at least rework the text so it is a bit more careful: right now the authors provide very limited data on off-target effects - and given the potential membrane-disrupting effects of the TPP group this is a shame. The authors are potentially missing out on a non-trivial story here.
Reviewer 2 Report
Manuscript by Chen & Pavlova et al describes a new synthetic derivative of chloramphenicol (CHL), representing a beautiful example of how rational drug design may help improve the existing antibiotics. The authors continue their successful strategy of replacing dichloroacetyl moiety of CHL with different groups, such as histidyl- or lysyl- from their study from 2018. The new derivative includes an adaptable linker a TPP group aimed to make strong contacts with ribosomal RNA in the polypeptide tunnel. The authors use cell-based and biochemical assays to demonstrate that CAM-C4-TPP inhibits bacterial growth by interfering with protein synthesis. The efficiency of protein inhibition and ribosome binding is 3-5 times better than those of the parent antibiotic CHL. The authors have determined a 2.8A crystal structure of a 70S ribosome with CAM-C4-TPP, in which unbiased electron density resolves the new antibiotic in the PTC and tunnel. The structure nicely rationalizes how CAM-C4-TPP interacts with rRNA, resulting in improved binding and inhibition. Moreover, the authors use toe-printing to demonstrate that CAM-C4-TPP strongly stalls ribosomes at the initiation codon, and at the subsequent Ala and Ser residues, echoing the selective mode of action of CHL. The discussion throughout the manuscript is balanced, and conclusions are supported by the data.
In sum, this is a thorough, well-presented and beautifully illustrated study that deserves publication after addressing a few minor comments:
- In the abstract and elsewhere, replace the word “idiosyncratic” with a more mechanistic/structural term that better explains the mechanism of action.
- Explain why expression of RFP results in green color for LEV (in the text and/or Fig. 2A).
- On page 6, the authors hypothesize that C4-TPP may act via “e.g., membrane damage”. Please cite relevant literature showing that similar compounds act via this mechanism.
- Residue A2503 is mislabeled in Figure 5, panels A, E and F.
- This is up to the authors: the readers may appreciate it if the extensive comparison of various inhibitors and their binding modes is supplemented by a table with calculated buried surfaces areas.
